# Odontogenic Maxillary Sinusitis: Therapeutic Management of Cases with Oroantral Fistulae

Yasutaka Yun [1,2,*], Masao Yagi [1], Tomofumi Sakagami [1], Shunsuke Sawada [3], Yuka Kojima [3], Tomoe Nakatani [4], Risaki Kawachi [1], Kensuke Suzuki [1], Hideyuki Murata [1], Akira Kanda [1], Mikiya Asako [1] and Hiroshi Iwai [1]

1. Department of Otorhinolaryngology, Head & Neck Surgery, Kansai Medical University, Osaka 573-1010, Japan; yagimas@hirakata.kmu.ac.jp (M.Y.); sakagato@hirakata.kmu.ac.jp (T.S.); kawachir@hirakata.kmu.ac.jp (R.K.); suzukken@hirakata.kmu.ac.jp (K.S.); muratah@hirakata.kmu.ac.jp (H.M.); akanda@hirakata.kmu.ac.jp (A.K.); asako@hirakata.kmu.ac.jp (M.A.); iwai@hirakata.kmu.ac.jp (H.I.)
2. Department of Otorhinolaryngology, Takeda General Hospital, Kyoto 601-1495, Japan
3. Department of Oral and Maxillofacial Surgery, Kansai Medical University, Osaka 573-1010, Japan; sawadash@hirakata.kmu.ac.jp (S.S.); kojimayk@hirakata.kmu.ac.jp (Y.K.)
4. Department of Oral and Maxillofacial Surgery, Takeda General Hospital, Kyoto 601-1495, Japan; teehaist@yahoo.co.jp
* Correspondence: yunys@hirakata.kmu.ac.jp; Tel.: +81-72-804-0101; Fax: +81-072-804-2547

**Abstract:** Odontogenic maxillary sinusitis (OMS) is a disease in which inflammation from the teeth extend into the maxillary sinus, causing symptoms of unilateral sinusitis. OMS can recur, with some being resistant to antibiotics. In intractable cases, exodontia and endoscopic sinus surgery (ESS) are necessary treatments. Here we report our analysis on the indications for surgical intervention in cases diagnosed with and treated as OMS. We retrospectively examined 186 patients who were diagnosed with sinusitis on a computed tomography (CT) scan. For cases diagnosed with OMS, the site of the causative tooth and the presence or absence of oroantral fistula to the maxillary sinus was examined. In addition, we analyzed the therapeutic efficacy of the initial treatment of antibiotics, and what the indications were for ESS. Among the patients examined, OMS was diagnosed in 44 cases (23.6%). In 14 out of 20 cases that underwent a post-medical treatment CT scan, OMS found to be treatment-resistant. Of these 14 cases, 12 (88%) had oroantral fistulae to the maxillary sinus. In all cases where exodontia, fistula closure surgery, and endoscopic sinus surgery (ESS) were performed, the fistula disappeared and the shadow of inflammation in the paranasal sinus improved. In OMS with oroantral fistula, ESS, exodontia, and fistula closure should be recommended over medication such as macrolide therapy.

**Keywords:** odontogenic maxillary sinusitis; macrolide therapy; oroantral fistula; endoscopic sinus surgery; exodontia

## 1. Introduction

Odontogenic maxillary sinusitis (OMS) is a disease in which inflammation of the teeth extend into the maxillary sinus, causing symptoms of unilateral sinusitis (e.g., nasal discharge, facial pain, foul odor, cacosmia, etc.) [1,2]. OMS is a common condition encountered in daily practice. Depending on the severity of inflammation, pan-sinusitis and other serious conditions such as nasal, intracranial, and orbital complications may occur [3,4]. The most common cause is the presence of apical periodontitis due to untreated caries or an inadequately treated root canal [5,6]. Apical periodontitis, also referred to as apical root lesion, is caused by inflammation spreading to the apex of tooth. This is further classified into pyogenic periodontitis (Figure 1A), in which inflammation expands through the root canal, and marginal periodontitis (Figure 1B), in which inflammation expands through the periapical space [5]. Bone resorption in the alveolar bone of the maxillary sinus due to inflammation may lead to traffic in the maxillary sinus, resulting in oroantral fistula

(Figure 1C,D). These fistulae are defined as an unnatural communication between the oral cavity and maxillary sinus with epithelialization in the fistula tract [7]. Furthermore, Felisati et al. reported that OMS can be classified into three groups depending on the etiology of the complication. Groups 1 and 2 are defined as sinusitis caused by the treatment of implants, while Group 3 is defined caused by classic dental disease such as caries or dental treatment complications [8]. The treatments of choice are antibacterial drugs and dental treatment of the tooth, similar to how sinusitis is managed [9]. However, conditions that require treatment by an otorhinolaryngologist and dentist must often be carried out at the same time, which may lead to delays in treatment due to poor coordination. In addition, the disease may be resistant to antibacterial therapy and may have recurrent symptoms [2]. Refractory cases should be treated with dental procedures, such as exodontia, or surgical treatment, such as endoscopic sinus surgery (ESS) to open the sinuses, although there is a lack of consensus on the indications for surgical treatment [6,9].

Here, we report on the evaluation of the indications for surgery in previously treated OMS.

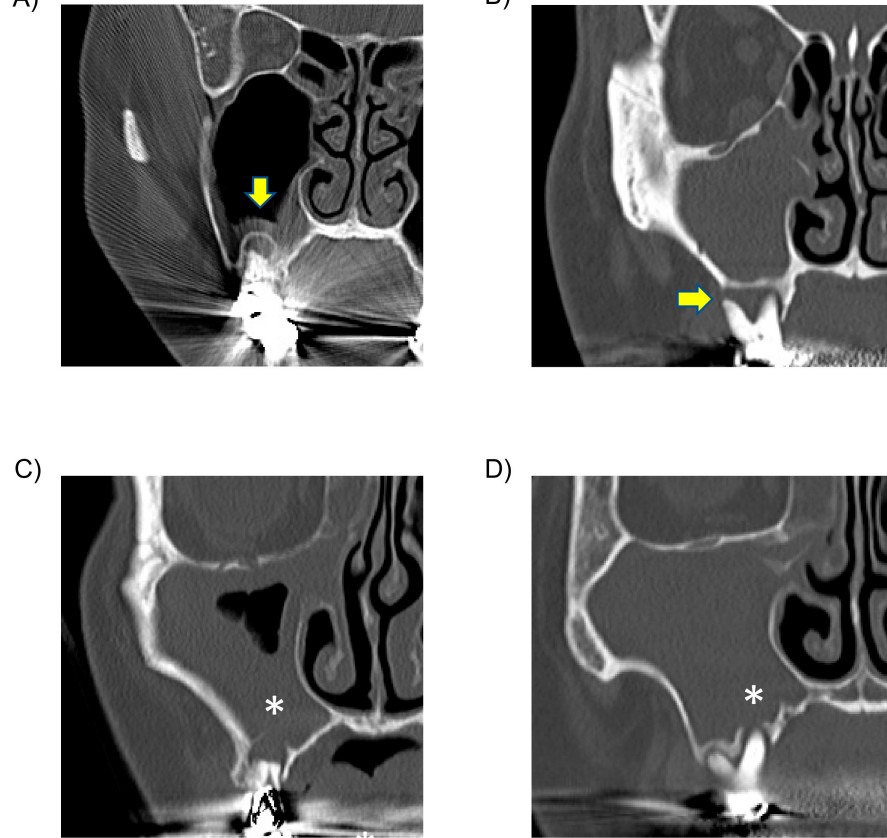

**Figure 1.** (**A**) Pyogenic periodontitis of the roots. (Case) A 46-year-old woman. A cystic lesion was found on the root apex of the right upper first molar. The yellow arrow points to the root apex lesion; (**B**) Marginal periodontitis. (Case) A 40-year-old woman. There is bone resorption around the alveolar bone of the right upper first molar. A yellow arrow points to the root apex lesion. (**C**) Pyogenic radicular periodontitis. Fistulae into the maxillary sinus are found in the area indicated by the asterisk. (**D**) Marginal periodontitis. (Case) A 68-year-old woman. A fistula into the maxillary sinus was found in the right upper first molar as indicated by the asterisk.

## 2. Material and Methods

We retrospectively reviewed the sinus computed tomography (CT) findings of 186 patients who were seen at our clinic from April 2015 to March 2016 (12 months) and diagnosed with sinusitis, including OMS. Sinus CT findings were evaluated in detail using horizontal sections, coronal sections, and sagittal sections. We diagnosed OMS by the presence of soft

shadows in the maxillary sinus and ipsilateral root lesions [5]. We excluded cases of sinus mycosis, eosinophilic chronic sinusitis, allergic fungal sinusitis, postoperative maxillary sinus cysts, and cases of foreign bodies such as dental materials and implants. All cases were classified as Group 3 in the OMS classification [8]. Next, root lesions were assessed in cases of diagnosed OMS. Patients asked to provide information about their age and sex, and the location of the causative tooth and the presence of oroantral fistulae were examined in by an otorhinolaryngologist and dentist. We also evaluated the efficacy of initial antibacterial treatment and the indication for subsequent surgery in cases of OMS.

We evaluated group differences for statistical significance using Fisher's exact test. We considered all *p* values <0.05 as statistically significant for all tests. All statistical analyses were performed using GraphPad Prism ver.9 (GraphPad Software, San Diego, CA, USA).

The local ethics committees at the Takeda general hospital (No 2020-019) approved this study.

## 3. Results

Of the 186 patients diagnosed with sinusitis, 44 were diagnosed with OMS. The mean age of the patients was $50.0 \pm 15.0$ years, and the male-to-female ratio was approximately 1:1. The age distribution showed bimodality in the 40s and 60s (Figure 2A).

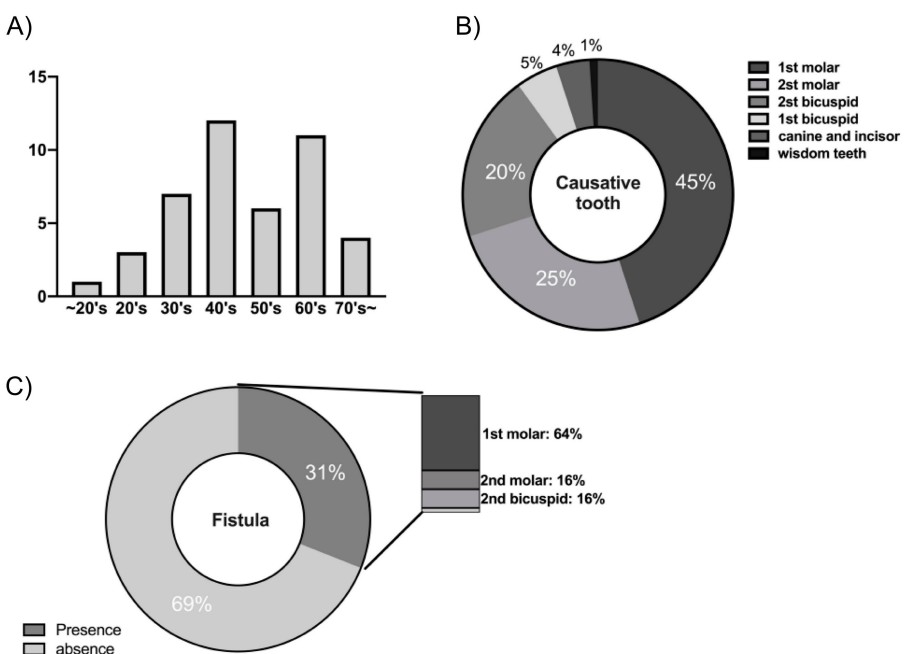

**Figure 2.** (**A**) Age distribution. Average age is $50.0 \pm 15.0$ years old, male to female ratio is about 1:1; (**B**) Causative tooth details; (**C**) Presence and location of oroantral fistulae of all root apex lesions.

Sinus CT findings showed soft shadows in the maxillary sinus only in 11 cases (25%), while the other 33 cases (75%) had soft shadows in multiple sinuses. The diseased side was on the right in 24 cases (54%), on the left in 11 cases (25%), and on both sides in 9 cases (21%). When looking at the location of the root apex lesions, 25 (56%) had lesions on a single root, 11 (25%) on 2 roots, and 8 (19%) on 3 or more roots. The causative tooth was found on 76 roots in all cases, with the most common cause being the first molar (sixth) on both sides. The total number of roots included 3 roots (4%) of canines and incisors, 4 roots (5%) of the first bicuspid, 15 roots (20%) of the second bicuspid, 34 roots (45%) of first molars, 19 roots (25%) of second molars, and 1 root (1%) of a wisdom tooth (Figure 2B).

Resorption of the alveolar bone of the maxillary sinus and suspected oroantral fistulae with the maxillary sinus were found in 23 of 44 cases (52%) and in 24 roots (31%) of all root apex lesions. The total number of roots included 1 root (4%) in the first bicuspid, 4 roots

(16%) in the second bicuspid, 15 roots (64%) in the first molar, and 4 roots (16%) in the second molar (Figure 2C).

After the diagnosis of OMS, the most common treatment option was intranasal corticosteroid (Fluticasone furoate) and macrolide therapy (clarithromycin, 200 mg/day) in 32 cases (72%), followed by 10 cases (23%) that were initially prescribed fluoroquinolones according to the treatment of acute sinusitis then shifted to macrolides (clarithromycin, 200 mg/day) [10].

We studied 20 patients who were treated with macrolides for more than 1 month (mean 2.8 ± 1.1 months) and who were able to undergo repeat sinus CT (Table S1). Of the 20 cases, 6 cases (30%) were in the "treatment-responsive" group because of the disappearance of the sinus soft shadow after macrolide therapy. On the other hand, 14 patients (70%) were considered to be "treatment-resistant" with residual or unchanged sinus soft shadows (Figure 3A). Next, the presence of maxillary sinus fistulae due to alveolar bone resorption was assessed. Fistulae were found in 2 of 6 patients (33%) in the treatment-responsive group and in 12 of 14 patients (88%) in the treatment-resistant group, indicating that the patients with oroantral fistulae were significantly more resistant to treatment (Figure 3B).

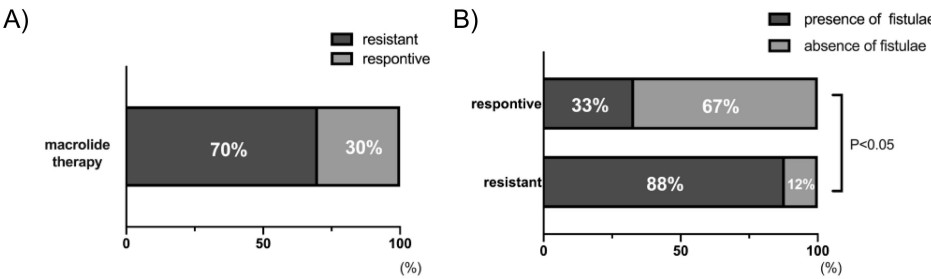

**Figure 3.** Outcome and prognostic factors after macrolide therapy in patients with OMS. (**A**) Outcome after macrolide therapy; (**B**) outcome by the presence or absence of oroantral fistulae. $p < 0.05$. Fisher's exact test.

After macrolide therapy, 11 of 20 patients underwent simultaneous exodontia of the causative tooth, fistula closure with a buccal flap [11], and opening of the sinus by ESS. Improvement of sinusitis and disappearance of oroantral fistulae were noted in all cases wherein macrolide therapy was continued after surgery [10]. Case No. 7 did not have a fistula at the floor of the maxillary sinus but was refractory to treatment. Thus, this patient underwent ESS only with noted improvement of symptoms of sinusitis postoperatively. Case No. 8 showed a fistula at the base of the maxillary sinus and was refractory to treatment. The patient was treated initially with ESS for sinusitis but the sinusitis did not improve much postoperatively. Therefore, exodontia was performed at a later date, and the sinusitis was improved. Case No. 12 had a fistula at the base of the maxillary sinus and was refractory to treatment. The patient had an exodontia operation in the outpatient clinic but the postoperative sinus symptoms remained. Therefore, ESS was performed, and the sinusitis improved. Case No. 20 did not have a fistula at the base of the maxillary sinus but was refractory to treatment. At the decision of the dentist, the patient underwent a prior outpatient exodontia. However, the patient still had sinus shadow on the CT and is under observation.

## 4. Discussion

Previous reports have shown that 10–40% of chronic sinusitis cases are caused by OMS with root tip lesions [2,12,13]. In the present study, we classified and evaluated periodontitis according to the presence or absence of oroantral fistulae.

It is important to identify the causative teeth and evaluate the extent of inflammation and sinusoidal shadowing to determine a treatment strategy for OMS. Traditionally, simple radiographic examinations such as the Waters method and dental panoramic imaging have been used. However, these tomographic scans are two-dimensional, and the presence of

artifacts makes diagnosis difficult [9]. As a result, there is difficulty in evaluating bone resorption in the alveolar bone of the maxillary sinus and in confirming the extent of root lesions and the presence of oroantral fistulae. In recent years, however, CT imaging has become widespread enough to generate images in multiple directions using sagittal and coronal sections in addition to horizontal sections. As a result, CT imaging is said to be the most useful in the diagnosis of root apex lesions and alveolar bone resorption in the maxillary sinus [9,14]. Cone-Beam CT is becoming more common in the field of otolaryngology due to its high spatial resolution, low radiation dose, and reduced effect of metal artifacts, which makes it possible to evaluate apex lesions in detail [15,16].

When looking at the teeth responsible for our department's cases of OMS, the most common site was the first molar, followed by the second molar. These results were reported by other authors as well [17]. Oroantral fistula is among the most common cases of OMS, accounting for approximately 60% of the cases, which is similar to our results [7]. This may be attributed to the proximity and short distance between the root apex lesion and the base of the maxillary sinus, which can easily form a fistula to bone resorption of the alveolar bone of the maxillary sinus [1].

Macrolide therapy is recommended as one of the medications for chronic sinusitis, including OMS. The main effects of macrolide drugs, including their ameliorative effects on the mucus-fibrillar transport system and inhibition of proinflammatory cytokines, have been reported to improve sinusitis in approximately 90% or more after 3 months of medication [18]. However, our results showed that there was no significant improvement in 70% of all cases of OMS, and 88% of cases with a fistula into the maxillary sinus were refractory to treatment. The reasons for treatment resistance may be due to inflammation caused by bacteria and infectious agents as well as severe damage to the adjacent maxillary sinus mucosa. Treatment-resistant OMS must be managed instead with radical treatment of the causative tooth and treatment of the sinusitis [19].

Our evaluation of cases in which exodontia and fistula closure were performed concurrently with ESS showed that there was an improvement in sinusitis in all cases based on postoperative sinus CT. This suggests that the removal of the infected root lesion and closure of the fistula, as well ESS, improves sinus ventilation and drainage.

In cases of OMS with oroantral fistulae, we recommend surgical treatment with ESS, exodontia, and fistula closure over medication such as macrolide therapy.

## 5. Conclusions

Resistance of medication such as a long-term macrolide therapy for OMS is associated with the presence of oroantral fistulae. For such patients, surgical treatment with ESS, exodontia, and fistula closure should be recommended over macrolide therapy.

**Supplementary Materials:** The following are available online at https://www.mdpi.com/2309-107 X/5/1/6/s1, Table S1: Detail Character of Patient after Macrolide Therapy.

**Author Contributions:** Conceived and designed the experiments: Y.Y. Performed the methodology and surgery: Y.Y., M.Y., T.S., T.N., S.S., Y.K., K.S., R.K., H.M., A.K., M.A. Analyzed the data: Y.Y., and H.I. Wrote the paper: Y.Y., and H.I. All authors have read and agreed to the published version of the manuscript.

**Funding:** This research was supported by grants from the Fund of Academic Society for Research in Otolaryngology, Kansai Medical University.

**Institutional Review Board Statement:** The study was conducted according to the guidelines of the Declaration of Helsinki, and approved by Ethics Committee of Takeda general hospital (No.2020-019).

**Informed Consent Statement:** Informed consent was obtained from all subjects involved in the study.

**Data Availability Statement:** The data presented in this study are available on request from the corresponding author.

**Conflicts of Interest:** The authors have no conflicts of interest to declare.

## Abbreviations

OMS: odontogenic maxillary sinusitis, ESS: endoscopic sinus surgery, CT: computed tomography.

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
