# Peer review of "Odontogenic Maxillary Sinusitis: Therapeutic Management of Cases with Oroantral Fistulae"

_2673-351X, doi:10.3390/sinusitis5010006_

Round 1

Reviewer 1 Report

Authors made a retrospective review of patients with OMS focusing on cases with apical root lesions with particular attention to those with communication within the sinus.

The main weaknesses of the study are its retrospective nature, a low number of cases, and the lack of a management protocol related to the patient’s signs and symptoms.

No mention of the patient’s symptoms has been done in the description of cases. Symptomatology could be one determining factor to decide whether to surgically operate in a short time or not. Since orbital and intracranial complications are rare an early intervention should more be related to the patient’s symptoms rather than the risk of complications. These evaluations should be added to the text.

The title is too generic since the authors mainly report OMS due to apical periodontitis.

The authors should also refer to any classification of OMS proposed by other authors to better define the pathology treated (for example Odontogenic sinusitis and sinonasal complications of dental treatments: a retrospective case series of 480 patients with critical assessment of the current classification. Marco Molteni. Acta Otorhinolaryngol Ital. 2020 Aug;40(4):282-289.  doi: 10.14639/0392-100X-N0457)

Results should be reviewed. In line 94, it is written that 11 patients underwent ESS+exodontia but in the following line and in the table, they appear to be only 10.

Why did the authors decide to treat with exodontia/ESS some patients responsive to the treatment in some cases and not in other cases?

Why case 5 did not undergo any surgical therapy?

How the authors managed cases not responsive after surgical treatment such as cases 8-12?

What about case 11? All cases which underwent only ESS or only exodontia have been described in detail except case 11.

What about bilateral cases? The authors report 9/44 bilateral cases, none of them was among the selected 20 patients?

Line 96, how long was medical therapy continued after surgery? Usually, after a combined therapy, there is no need to continue an antibiotic therapy after surgery…do the authors have any protocol for medical therapy after surgical treatment?

The term “fistula” used in the text is not clear and should be better defined. Is it used to mean also an oroantral communication or only an opening of a periapical cyst into the sinus? Of course, in absence of an oroantral communication with epithelialization of the fistula tract (as seems to be for the cases reported) the % of success after surgery is higher, and also the treatment would be different (closure of the fistula with a flap). How did exactly the “fistula closure” has been performed? Was an oroantral fistula present in the cases reported?

No information about the statistical analysis has been reported in the material and methods section

Line 125 is not clear if 20% of patients had or didn’t have a fistula (the original article is only in Japanese). Authors reported 52% of fistulas which are higher than 20%

What about steroid therapy? It could be added to antibiotics therapy. Did any patients undergo steroid therapy for sinusitis?

Medical therapy with Macrolides could be effective for chronic rhinosinusitis but seems not to be a gold standard for treatment of odontogenic sinusitis or with unilateral disease and three months of therapy seems to be excessive especially when considering “early surgery” [Odontogenic sinusitis: developments in diagnosis, microbiology, and treatment. Workman AD, et al. Curr Opin Otolaryngol Head Neck Surg. 2018] and, as already stated by different authors, “For endodontic disease, antibiotic therapy without definitive dental treatment has been considered inappropriate.” [Management of odontogenic sinusitis: multidisciplinary consensus statement John R Craig et al Int Forum Allergy Rhinol 2020 Jul;10(7):901-912.  doi: 10.1002/alr.22598. Epub 2020 Jun 7.]. This could create a strong bias in the final results.

It is not clear at the end of the article what the authors mean for early intervention since they stated that patients should be treated once they have been determined to be treatment-resistant.

Limitations of the study should be added (small sample size, retrospective nature, etc.)

Minor corrections:

Line 86: modify 20 with Twenty

Table S1: responsive instead of respontive

Remove figure “2C” and figure 3 which does not add anything more to the text

Author Response

To reviewer 1

Major comments 

  1. No mention of the patient’s symptoms has been done in the description of cases. Symptomatology could be one determining factor to decide whether to surgically operate in a short time or not. Since orbital and intracranial complications are rare an early intervention should more be related to the patient’s symptoms rather than the risk of complications. These evaluations should be added to the text.

Answer: Thank you for your comment. The patient’s symptoms are very important to decide whether to surgically operation. In our results, there were some cases in which macrolide therapy improved symptoms, but imaging findings showed no change. Therefore, we determined that surgery should be based on the presence or absence of oroantral fistula rather than symptoms.

  1. The title is too generic since the authors mainly report OMS due to apical periodontitis.

Answer: According to reviewer’s suggestion, we changed the titles.

  1. The authors should also refer to any classification of OMS proposed by other authors to better define the pathology treated (for example Odontogenic sinusitis and sinonasal complications of dental treatments: a retrospective case series of 480 patients with critical assessment of the current classification. Marco Molteni. Acta Otorhinolaryngol Ital. 2020 Aug;40(4):282-289. doi: 10.14639/0392-100X-N0457)

Answer: We agree with your comments. We cited this article and used this classification of OMS 1. Please see the added sentences in line 53-56 and line 72.

  1. Results should be reviewed. In line 94, it is written that 11 patients underwent ESS+exodontia but in the following line and in the table, they appear to be only 10.

Answer: The description in this tableS1 was incorrect. We have corrected.

  1. Why did the authors decide to treat with exodontia/ESS some patients responsive to the treatment in some cases and not in other cases?

Answer: Thank you for your comment. Cases No.2 and No.18 were in the treatment-resposive group, but ESS and exodontia were performed at the patient's request.

  1. Why case 5 did not undergo any surgical therapy?

Answer: No.5 case patient cancelled just before the surgery.

  1. How the authors managed cases not responsive after surgical treatment such as cases 8-12?

Answer: No.8 patient, ESS was performed first, but there was little improvement. Therefore, exodontia was performed at a later date, and the patient improved. In other hand,

No.12 patient exodontia was performed first, ESS was performed at a later date, and the patient improved. We modified sentences in line 116, and line 118-119.

  1. What about case 11? All cases which underwent only ESS or only exodontia have been described in detail except case 11.

Answer: It is corrected. Please see comment about No.4 answer.

  1. What about bilateral cases? The authors report 9/44 bilateral cases, none of them was among the selected 20 patients?

Answer: Thank you for your comment. There were 5 patients has bilateral sinusitis.

  1. Line 96, how long was medical therapy continued after surgery? Usually, after a combined therapy, there is no need to continue an antibiotic therapy after surgery…do the authors have any protocol for medical therapy after surgical treatment?

Answer: As for macrolide therapy after ESS, postoperative continuation of macrolide therapy is performed according to this manuscript at our hospital. 2. We have cited this manuscript.

  1. The term “fistula” used in the text is not clear and should be better defined. Is it used to mean also an oroantral communication or only an opening of a periapical cyst into the sinus? Of course, in absence of an oroantral communication with epithelialization of the fistula tract (as seems to be for the cases reported) the % of success after surgery is higher, and also the treatment would be different (closure of the fistula with a flap). How did exactly the “fistula closure” has been performed? Was an oroantral fistula present in the cases reported?

Answer: We agree with your comment. We added sentences detailing the definition of fistulae and fistula closure in the line 51-56. And we changed “fistula” to “oroantral fistula”. Because, in our case, it was diagnosed by the dental doctor as oroantral fistula, not periapical cyst opening. 3,4 5 We added a sentence how to close the fistulae 4.

  1. No information about the statistical analysis has been reported in the material and methods section

Answer: We added the method about statistical analysis in the line 76-78.

  1. Line 125 is not clear if 20% of patients had or didn’t have a fistula (the original article is only in Japanese). Authors reported 52% of fistulas which are higher than 20%

Answer: Please see comment No.11.

14.What about steroid therapy? It could be added to antibiotics therapy. Did any patients undergo steroid therapy for sinusitis?

Answer: According to reviewer’s suggestion, we added a sentence in line 96-97.

  1. Medical therapy with Macrolides could be effective for chronic rhinosinusitis but seems not to be a gold standard for treatment of odontogenic sinusitis or with unilateral disease and three months of therapy seems to be excessive especially when considering “early surgery” [Odontogenic sinusitis: developments in diagnosis, microbiology, and treatment. Workman AD, et al. Curr Opin Otolaryngol Head Neck Surg. 2018] and, as already stated by different authors, “For endodontic disease, antibiotic therapy without definitive dental treatment has been considered inappropriate.” [Management of odontogenic sinusitis: multidisciplinary consensus statement John R Craig et al Int Forum Allergy Rhinol 2020 Jul;10(7):901-912. doi: 10.1002/alr.22598. Epub 2020 Jun 7.]. This could create a strong bias in the final results.

Answer: We fully agree your comment. OMS was resistant to antibiotic therapy such as long-term macrolide therapy, and our results showed resistance especially in cases with oroantral fistulae. For this reason, we concluded that surgical treatment is recommended over medication in cases with oroantral fistula. We modified sentences in line 158-159.

  1. It is not clear at the end of the article what the authors mean for early intervention since they stated that patients should be treated once they have been determined to be treatment-resistant.

Answer: Thank you for your comment. Please see answer No.14. We modified conclusion in the line 161-163.

  1. Limitations of the study should be added (small sample size, retrospective nature, etc.)

Answer: Thank you for your comment. Please see line 66.

Minor comment

18.Line 86: modify 20 with Twenty.

Answer: It is corrected in the line 100.

19.Table S1: responsive instead of respontive.

Answer: It is corrected in Table S1.

20.Remove figure “2C” and figure 3 which does not add anything more to the text.
Answer: Thank you for your comment. But figure2C and 3 is very important for this manuscript. We can't agree to delete the figure.

Reviewer 2 Report

This is a clinically relevant topic, as the entity of OMS is likely underdiagnosed and underappreciated by clinicians.

The authors seem primarily concerned with treatment of patients with a fistula associated with OMS.   The findings do not directly support the conclusion, "Early intervention with, exodontia, and fistula closure should be
immediately performed."  Note that 33% of the treatment-responsive group also had fistula, yet did not require surgery.  The authors should temper their conclusions.

Similarly, the results do not support the conclusion, "There is a high risk of treatment failure when conservatively treating patients with OMS who
have a fistula in the maxillary sinus."  The analysis does not determine risk.  They could instead conclude that in this limited sample of patients, failure of antibiotic therapy for OMS is associated with the presence of a fistula.

I would suggest distinguishing chronic from acute sinusitis in the methods and results.

The duration of macrolide therapy should be specified, as well as the name and dosing of the drug.

The distribution of involvement of specific teeth is reported.  Is there significance to OMS arising from one tooth versus another?

Author Response

To reviewer2

1.This is a clinically relevant topic, as the entity of OMS is likely underdiagnosed and underappreciated by clinicians.

The authors seem primarily concerned with treatment of patients with a fistula associated with OMS.   The findings do not directly support the conclusion, "Early intervention with, exodontia, and fistula closure should be immediately performed."  Note that 33% of the treatment-responsive group also had fistula, yet did not require surgery.  The authors should temper their conclusions.

Answer: Thank you for your comment. We agree your opinion. We modified sentences in conclusions (Line 158-163).

2.Similarly, the results do not support the conclusion, "There is a high risk of treatment failure when conservatively treating patients with OMS who have a fistula in the maxillary sinus."  The analysis does not determine risk.  They could instead conclude that in this limited sample of patients, failure of antibiotic therapy for OMS is associated with the presence of a fistula.

Answer: We modified sentences in conclusions (Line 158-163)

3.I would suggest distinguishing chronic from acute sinusitis in the methods and results.

The duration of macrolide therapy should be specified, as well as the name and dosing of the drug.

Answer: Thank you for your suggestion. We add a sentence about detail of macrolide therapy in line 97-99.

4.The distribution of involvement of specific teeth is reported.  Is there significance to OMS arising from one tooth versus another?

Answer: Thank you for comment. There are reports that first molars are more frequently caused by OMS. Please see sentences in the line 142-144.

  1. Molteni M, Bulfamante AM, Pipolo C, et al. Odontogenic sinusitis and sinonasal complications of dental treatments: a retrospective case series of 480 patients with critical assessment of the current classification. Acta Otorhinolaryngol Ital 2020;40:282-9.
  2. Nakamura Y, Suzuki M, Yokota M, et al. Optimal duration of macrolide treatment for chronic sinusitis after endoscopic sinus surgery. Auris Nasus Larynx 2013;40:366-72.
  3. Dym H, Wolf JC. Oroantral Communication. Oral and Maxillofacial Surgery Clinics of North America 2012;24:239-47.
  4. Borgonovo AE, Berardinelli FV, Favale M, Maiorana C. Surgical options in oroantral fistula treatment. Open Dent J 2012;6:94-8.
  5. Galli M, De Soccio G, Cialente F, et al. Chronic maxillary sinusitis of dental origin and oroantral fistula: The results of combined surgical approach in an Italian university hospital. Bosn J Basic Med Sci 2020;20:524-30.

Round 2

Reviewer 1 Report

Please correct minor changes before pubblication:

Modify line 22-24 as follow: “Among the patients examined in 44 cases (23.6%) an OMS was diagnosed and 14 out of 20, who performed a post-medical treatment CT scan, were found to be treatment resistant.”

Line 52 “as” instead of “that”

Line 111. What “(10)” stands for? It has been corrected in table but not in the text. Remove or correct  “(10)”.

Line 159 Such as instead of such “us”

Author Response

Date Jun 4,2021

Dear Dr. Eleonora Nucera

Dear Rerviewers

Manuscript ID:  sinusitis-995515

Type of manuscript:  Original article

Title:  Revision of manuscript titled " Odontogenic Maxillary Sinusitis: Therapeutic management of cases with oroantral fistulae."

Authors: Yasutaka Yun, Masao Yagi, Tomofumi Sakagami, Shunsuke Sawada, Yuka Kojima, Tomoe Nakatani, Risaki Kawachi, Kensuke Suzuki, Hideyuki Murata, Akira Kanda, Mikiya Asako, and Hiroshi Iwai

We would like to thank for Reviewer’s comments and valuable suggestions on our manuscript. Based on suggestions, we revised our manuscript.

We tried to correct the text as commented by Reviewer 1.(modifications in the text are indicated by red).

Reviewer 2 Report

Thank you for addressing my concerns.

Author Response

Date Jun 4,2021

Dear Dr. Eleonora Nucera

Dear Rerviewers

Manuscript ID:  sinusitis-995515

Type of manuscript:  Original article

Title:  Revision of manuscript titled " Odontogenic Maxillary Sinusitis: Therapeutic management of cases with oroantral fistulae."

Authors: Yasutaka Yun, Masao Yagi, Tomofumi Sakagami, Shunsuke Sawada, Yuka Kojima, Tomoe Nakatani, Risaki Kawachi, Kensuke Suzuki, Hideyuki Murata, Akira Kanda, Mikiya Asako, and Hiroshi Iwai

We would like to thank for Reviewer’s comments and valuable suggestions on our manuscript. Based on suggestions, we revised our manuscript.